# Dopamine Transmission Imbalance in Neuroinflammation: Perspectives on Long-Term COVID-19

**DOI:** 10.3390/ijms24065618

**Published:** 2023-03-15

**Authors:** Maria Mancini, Silvia Natoli, Fabrizio Gardoni, Monica Di Luca, Antonio Pisani

**Affiliations:** 1Department of Brain and Behavioral Sciences, University of Pavia, 27100 Pavia, Italy; maria.mancini@unipv.it; 2IRCCS Mondino Foundation, 27100 Pavia, Italy; 3Department of Clinical Science and Translational Medicine, University of Rome Tor Vergata, 00133 Rome, Italy; silvia.natoli@uniroma2.it; 4IRCCS Maugeri Pavia, 27100 Pavia, Italy; 5Department of Pharmacological and Biomolecular Sciences “Rodolfo Paoletti”, University of Milan, 20133 Milan, Italy; fabrizio.gardoni@unimi.it (F.G.); monica.diluca@unimi.it (M.D.L.)

**Keywords:** dopamine, SARS-CoV-2, neuroinflammation, cytokines, Parkinson’s disease, dopamine release, interleukins, glia, alpha-synuclein, long-COVID, post-acute sequelae

## Abstract

Dopamine (DA) is a key neurotransmitter in the basal ganglia, implicated in the control of movement and motivation. Alteration of DA levels is central in Parkinson’s disease (PD), a common neurodegenerative disorder characterized by motor and non-motor manifestations and deposition of alpha-synuclein (α-syn) aggregates. Previous studies have hypothesized a link between PD and viral infections. Indeed, different cases of parkinsonism have been reported following COVID-19. However, whether SARS-CoV-2 may trigger a neurodegenerative process is still a matter of debate. Interestingly, evidence of brain inflammation has been described in postmortem samples of patients infected by SARS-CoV-2, which suggests immune-mediated mechanisms triggering the neurological sequelae. In this review, we discuss the role of proinflammatory molecules such as cytokines, chemokines, and oxygen reactive species in modulating DA homeostasis. Moreover, we review the existing literature on the possible mechanistic interplay between SARS-CoV-2-mediated neuroinflammation and nigrostriatal DAergic impairment, and the cross-talk with aberrant α-syn metabolism.

## 1. Introduction

Neurons transmit signals to each other at synapses through the release of neurotransmitters stored within the synaptic vesicles. Several different factors influence the neurotransmission process including the neurotransmitter availability and its rate-of-synthesis, the release, the number of postsynaptic receptors available for the neurotransmitter to bind to, the deactivation at the synaptic cleft level or the presynaptic reuptake and recycling. Changes at one or more of these steps alter synaptic communication, circuit functioning and plasticity with a powerful impact on behavior.

The brain neurochemistry can be perturbed by several stimuli, and dysregulation of the synapse biology with ensuing impairment of neurotransmission seems to be one of the leading effects of several viral and bacterial infections. Although the immune system and the central nervous system (CNS) have long been considered two distinct compartments, there is increasing evidence for a close communication between them. Indeed, the activation of the immune-neural-synaptic axis following immune challenge is mainly based on the ability of peripheral immune cells to infiltrate the brain [1] or contribute to the stimulation of glial cells through peripheral released humoral immune factors that can cross the blood brain barrier (BBB) [2]. Activation of glial cells causes further secretion of proinflammatory molecules, thereby engaging more immune cells to the CNS [3] and increasing the local inflammatory process.

The pattern of released proinflammatory molecules, such as cytokines, depends on the nature of the antigenic stimulus and the cell source that is being stimulated. Cytokine and chemokine receptors are distributed on most, if not all, brain cells. A given cytokine is able to modulate the expression and release of other cytokines in target cells, adding further complexity to the complete picture of stimuli and downstream signaling in neurons after a challenge with a single cytokine. These molecules can directly target neurons or can locally modulate synaptic transmission via glia-neuron signaling. At synaptic level, in fact, pre- and postsynaptic neuronal elements make connections with processes of the neighboring glial cells, astrocytes, and microglia, creating a multipartite synapse [4]. The binding of cytokines and chemokines to glial cells can determine the release of excitatory or inhibitory neuromodulators, which then signal to the neurons affecting the synaptic transmission [5,6]. Cytokines acting directly on neuronal elements without the glia as intermediary are still incompletely defined.

The neuromodulators released following cytokine receptor activation can affect several distinct steps of synaptic transmission, including the probability of neurotransmitter release, the postsynaptic sensitivity to neurotransmitter, and the membrane excitability of the pre- and postsynaptic cells.

The alteration of neurotransmission by inflammatory cytokines (e.g., interleukin (IL)1-β, IL-6, IL-8, tumor necrosis factor (TNF)-α, interferon (IFN)-α and IFN-γ) has been reported for several neurotransmitters, namely norepinephrine (NE), dopamine (DA), serotonin (5-HT), and glutamate [7].

DA is well known to control movement, motivation, and cognition, and perturbation of DA transmission is indeed implicated in the pathogenesis of a number of neuropsychiatric disorders, including Parkinson’s disease (PD), schizophrenia, and addiction. Recently, several reports have sought to elucidate the role of inflammation in these conditions and to uncover the cellular and molecular mechanisms underlying DA neurotransmission alterations. However, few studies have focused on the dysregulation of DA neurotransmission following infections.

Links between viral infections and PD have long been suspected and studied, but the exact relationship remains elusive. SARS-CoV-2 virus, the causative agent of COVID-19, does not cause evident neuropathological alterations in infected cells, such as necrotic changes or other cytological alterations [8]; however, a direct impact on the brain has been reported and seems to aggravate PD symptoms or trigger PD-like signs probably via an alteration of dopamine neurotransmission. In this sense, clinical data report describe a reduction in dopamine uptake in the putamen or in the striatum of people who have been infected with SARS-CoV-2 virus, which demonstrates nigrostriatal dysfunction [9,10]. Such cumulative damage seems to be dependent on an action of proinflammatory cytokines and chemokines on the mechanisms of dopamine release more than a direct viral effect on the substantia nigra.

In this review, we discuss recent reports of proinflammatory molecules, released during viral infections, that impact synaptic DA transmission in the mammalian CNS. Focus will be given to the infection caused by SARS-CoV-2 and the potential role of inflammatory/immune response in the pathogenesis of PD.

## 2. Mechanisms of DA Release and Regulation in the Nigrostriatal Pathway

Midbrain DA neurons in the substantia nigra pars compacta (SNc) and in the ventral tegmental area (VTA) provide the bulk of DA to the basal ganglia and the forebrain.

SNc DA neurons are perhaps the best studied because of their central role in the pathology of PD. Small in number, they send impressive axons to the forebrain, which densely ramify within the striatal region giving rise to the nigrostriatal DA pathway [11].

Both the axonal boutons in the forebrain and the somatodendritic compartment in midbrain release DA. The driving force for DA release is represented by the action potentials generated at the cell body level [12,13]. However, in some cases, DA release is independent of somatic action potential firing, and local striatal mechanisms (e.g., activation of β2-containing nicotinic acetylcholine receptors (nAChRs) on DA axons) lead to increases in extracellular DA concentration ([DA]_o_) [14]. Despite the existence of potential differences, both axonal and somatodendritic DA signaling are triggered by calcium (Ca^2+^) [15,16,17].

The release of DA and the amount of this neutrotrasmitter sensed by receptors is regulated at many stages including DA synthesis, vesicular transport, regulatory exocytotic proteins, and uptake, as well as Ca^2+^ homeostasis. These mechanisms are described in detail elsewhere [14,18,19]; only a brief overview is presented here (Figure 1).

DA production is mediated by the activity of tyrosine hydroxylase (TH), the rate-limiting enzyme of catecholamine synthesis. The activation of TH determines the hydroxylation of tyrosine to L-DOPA which, after decarboxylation to DA by the amino acid decarboxylase (AADC), is loaded into the synaptic vesicle. TH requires ferrous iron and tetrahydrobiopterin (BH4) as cofactors to perform the reaction (Figure 1). Post-translation modifications, such as phosphorylation reactions, influence the affinity for the cofactor BH4 causing changes in DA release [20].

The accumulation of DA into the synaptic vesicles is mediated by the vesicular monoamine transporter 2 (VMAT2) whose activity largely dictates quantal size affecting the scale of subsequent neurotransmitter release [21,22]. VMAT2 interacts with the DA synthesis machinery to sequestrate DA immediately after production (Figure 1). Since DA can be rapidly converted to reactive species within the oxidizing environment of the cytoplasm, this connection is important to prevent DA from damaging presynaptic terminals. DA uptake into striatal synaptic vesicles is decreased when interactions between VMAT2 with TH and AACD are lost or disrupted [23]. In addition, modifications in VMAT2 levels following changes in its synthesis or its relocalization [24,25,26] strongly correlate with increases or decreases in VMAT2-mediated DA uptake and synaptic vesicle DA content.

DA neurotransmission is initiated by fusion of synaptic vesicles in axonal boutons or the release from dendrites likely occurring via the fusion of specialized secretory organelles [27] (Figure 1). At axonal level, the force of fusion of synaptic vesicles with the presynaptic plasma membrane is provided by the formation of the SNARE complex between the vesicular (Synaptobrevin-2/VAMP-2) and the plasma membrane proteins (Syntaxin-1 and SNAP-25) [28] and the activation of Synaptotagmin 1, 2, or 9, which are the Ca^2+^ sensors that mediate vesicle fusion [29]. At the somatodendritic compartment level, different complements of SNARE proteins are found [30,31,32], suggesting alternative or additional mechanisms of DA release. In both cases, alterations in the SNARE complex formation or in the maintenance of its continuous assembly operated by alpha-synuclein (α-syn) protein [33] can modify the synaptic vesicle fusion event and thus impact DA release.

Once released from presynaptic terminals, DA mediates its effect by binding both the presynaptic and postsynaptic DA receptors belonging to the G-protein-coupled receptor (GPCR) family and termed D1 through D5 (Figure 1). These receptors, commonly segregated in the major classes D1-like receptors (D1 and D5) and D2-like receptors (D2, D3 and D4), differ in their structural, pharmacological, and signaling properties [34]. The D2-like receptors include also DA autoreceptors that play a crucial role in the regulation of axonal and somatodendritic DA release, DA neuron firing rate, and DA synthesis (Figure 1). Indeed, activation of D2 receptors in striatal slices causes a suppression of DA release [35,36]. Similarly, the binding of D2 receptors in midbrain by agonists inhibits somatodendritic DA release; at the soma, in fact, D2 receptors are coupled to G protein-gated inwardly rectifying potassium (GIRK) channels whose activation suppresses firing of DA neurons [37].

DA signaling is strongly influenced by DA clearance. While in midbrain, which has relatively sparse DA input, DA clearance is mainly dependent on diffusion and on the activity of NE transporter, in striatum, which is densely innervated by DA axons, DA clearance is mostly due to uptake by the DA transporter (DAT). As DA diffuses away from the synapse, DA action is termed by degradation or it is taken back into the presynaptic terminals via DAT, which is expressed perisynaptically only by DA neurons [38] (Figure 1). The packaging of DA into the synaptic vesicles is thus a mechanism used to restore the releasable pool of neurotransmitter; this recycling process provides a synthesis-independent strategy to refill synaptic vesicles. Changes in DAT expression or in its intracellular localization, and changes in DAT function caused by post-translational modifications and disruption in protein–protein interactions [39], can affect DA availability and DA release [40]. Indeed, DAT function can be regulated also by D2 receptor activation [41,42].

Mechanistic studies have revealed local regulation of DA release by transmitters such as glutamate, GABA, and acetylcholine (ACh) through direct or indirect mechanisms [14]. Similarly, neuroactive peptides and modulators coming from the periphery can modify [DA]_o_. Among these, inflammatory molecules released during disease states or during bacterial and viral infections have been reported to be able to alter several aspects of DA signaling, including synthesis, packaging, release, and reuptake, ultimately resulting in modifications of DA-dependent behaviors (Figure 1).

In the following sections we will analyze the effects of SARS-CoV-2 infection on DA release and the molecular mechanisms underlying the perturbations in DA levels.

## 3. SARS-CoV-2 Viral Infection and Dysregulation of DA Neurotransmission

### 3.1. Mechanisms of SARS-CoV-2 Invasion and Associated Immune Responses

SARS-CoV-2 is an enveloped, positive sense, single-stranded RNA virus. Its genome codes for the four structural proteins: the membrane, the envelope, the nucleocapsid, and the spike [43]. The latter, by binding the protein angiotensin-converting enzyme 2 (ACE2), and under the participation of type II transmembrane serine protease receptors (TMPRSS2) [44], allows the virus to enter the cell and replicate itself using the RNA and protein synthesis machinery of the host (Figure 2).

The recognition of pathogenic viral proteins by the body causes the activation of various peripheral immune cells that produce proinflammatory molecules, such as cytokines and chemokines, with the aim of helping the body to fight the virus. SARS-CoV-2 was demonstrated to activate several immune cells including macrophages/monocytes, T cells, neutrophils, and natural killer cells, able to kill the virus through cytokine release (Figure 2). Among the four proteins synthetized by SARS-CoV-2 virus, however, only the spike protein triggers the production of proinflammatory cytokines and chemokines [45].

The CNS is provided with a powerful barrier system limiting virus invasion. However, many viruses can invade the CNS, infect the neurons, propagate by exploiting cellular machinery, hijack axonal transport, and cross the synapses to disseminate within neural networks [46]. This is the case also for SARS-CoV-2, which exhibits regional neurotropism [47].

The trans-synaptic transfer across infected neurons is one way used by the virus to gain access to the CNS. Although the olfactory nerve and the ocular endothelium are considered the preferential routes of entry, a transneuronal retrograde transport via the lingual, the vagus, and the glossopharyngeal nerves can be used by the virus to invade the CNS after infection of the peripheral nerve terminals [48].

However, the neurovascular endothelium plays a crucial role in SARS-CoV-2 neuroinvasion. The ability of SARS-CoV-2 to infect endothelial cells causes dysfunction in BBB integrity [49] facilitating its access to the brain via infected peripheral immune cells. Indeed, the increased permeability of the BBB allows the peripheral inflammatory cytokines to gain access to the brain (Figure 1 and Figure 2).

Such invasion causes further secretion of various cytokines from the glial cells, astrocytes, and microglia, followed by the disruption of their homeostatic conditions, which indirectly impacts neuronal activities (Figure 2).

Analysis of human postmortem brain samples has revealed the presence of SARS-CoV-2 RNA in the striatum and in midbrain [50]. More interestingly, in a recent study, Emmi and colleagues [8], in addition to the genomic sequences, detected viral antigens within the substantia nigra. Immunoreactive neurons were found in proximity to blood vessels and expressed ACE2 and TMPRSS2 on their surface, suggesting the BBB and the structures of the peri- and juxtavascular compartment as the preferred route used by the virus to gain access to the substantia nigra. The affinity of SARS-CoV-2 for dopaminergic neurons has also been suggested by in vitro studies previously performed on cells and organoids from human pluripotent stem cells (hPSCs) [51,52], and by the expression of ACE2 receptor in the sustantia nigra of both human and mouse brains [53,54].

In addition to ACE2, other mechanisms for SARS-CoV-2 access to the brain have been identified. Neuropilin-1 (NRP1) [55], BASIGIN (BSG), Cathepsin L (CTSL) [56], furin [57], and CD147 [58] also facilitate SARS-CoV-2 to infect cells. All these proteins have a higher expression in the human brain, when compared to ACE2, and can be considered coreceptors favoring the virus entry into the CNS and its spread.

SARS-CoV-2 infection is associated with changes in functioning and responses of DA neurons. Despite the possibility by SARS-CoV-2 to infect neuronal cells, a direct effect of SARS-CoV-2 on DA cells has not been proven yet [8]. The clinical changes observed, potentially associated with PD pathogenesis [59,60,61], are rather caused by a robust inflammatory response characterized by massive cytokines production, persistent microglia activation, and abnormal α-syn levels, which alter DA neurotransmission and DA release during or after acute SARS-CoV-2 infection.

### 3.2. SARS-CoV-2-Induced Cytokine Storm

Cytokines are small pleiotropic proteins classically secreted by monocytes, macrophages, lymphocytes, and vascular endothelial cells in response to immune system challenge and able to reach the brain from the periphery following BBB disruption. Within neural tissues, the major source of cytokines is represented by microglial cells; however, other glial cells such as astrocytes can signal through cytokines also. Although neurons are typically considered a target for cytokines, they are a source of cytokines as well [62,63,64].

In healthy brains, several cytokines including ILs, IFNs, TNFs and chemokines have been identified. They have a wide range of functions, participating in synaptogenesis and synapse pruning, modulating synaptic plasticity, preserving neuronal homeostasis [65,66], and regulating neurotransmitter release.

The levels of cytokines are generally low at physiological-basal conditions; however, similar to that occurring in the peripheral tissues following viral infection, the cytokine brain levels dramatically increase in the face of an ensuing inflammatory or immune response. The inflammatory reaction is important for coping with CNS damage.

SARS-CoV-2 infection is often characterized by an excessive systemic production of proinflammatory cytokines known as “cytokine storm”. This poorly regulated release can temporarily impair or irreversibly damage tissue and organ functions [67]. Pathological inflammation and cytokine storm initiated by SARS-CoV-2 outside the brain also impact the CNS. Accordingly, analysis of cerebrospinal fluid (CSF) from patients with neurological symptoms found elevated levels of proinflammatory cytokines such as IL-1β, TNF-α, IL-6, IL-8 (CXCL8), IL-10, IL-15, IL-18, monocyte chemoattractant protein 1 (MCP-1), and macrophage inflammatory protein 1 alpha (MIP-1α) [68,69,70]. However, the neurological signs and symptoms associated with SARS-CoV-2 infection might not only depend on the increased CSF cytokines but also on serum cytokines.

Accumulating evidence points to dysregulation of neurotransmitter systems during SARS-CoV-2 infection, triggered by cytokines and proinflammatory molecules. The final downstream effects of cytokines on synaptic transmission depend on the synaptic cytokine concentrations, the balance between proinflammatory, and anti-inflammatory cytokines as well as the expression of their receptors [71].

### 3.3. Cytokine-Mediated Changes in Neurotransmission and DA Release

The localization of some cytokine receptors in the brain overlaps; however, some cytokine receptors have a precise distribution in some brain areas [63,72,73,74,75], suggesting the presence of region-specific functions.

Most of the studies performed at hippocampal and cortical level show that cytokines such as IL-1β, IL-6, IL-18, IFN-γ, TNF-α, TNF-β, and IL-17 influence synaptic transmission and plasticity during both physiological and pathological conditions [76,77,78,79,80,81,82,83,84,85,86]. Less is known about the effects of cytokines on synaptic transmission modulation in the subcortical areas. Within the basal ganglia circuitry, a regulation of synaptic transmission by cytokines has been reported. The frequency of presynaptic neurotransmitter release as well as the amplitude and the duration of the spontaneous postsynaptic currents, both excitatory and inhibitory (sEPSCs and sIPSCs), are significantly altered by these molecules [87,88,89,90,91,92,93].

Although most of the studies performed to investigate the effect of cytokines on synaptic transmission focused on excitatory and inhibitory neurotransmission, cytokines have also been proven to alter DA signaling. Indeed, brain DA has been consistently found to be modified by a variety of inflammatory stimuli in numerous studies performed to explore the effects of cytokines on healthy animals or brain slices.

DA levels are found to be impaired in several brain regions following administration of IL-1 [94]. In general, DA levels were increased by IL-1α [95,96] and IL-1β when administrated systemically [97]. In microdialysates from the nucleus accumbens, a mild enhancement induced by IL-1β was measured [97]. However, this observation is not confirmed in ex vivo studies. In fact, the evoked axonal DA release in ex vivo striatal slices of mice in which IL-1β is upregulated was reduced [98]. This result, corroborated by the evidence that the treatment with IL-1 receptor antagonist (IL-1ra) increases striatal DA levels [98], clearly demonstrates that this cytokine modifies DA neurotransmission. The differences in DA dynamics following IL-1β treatment, observed between in vivo and ex vivo studies, are probably due to the capacity of uptake from the blood to the brain that does not enable the same cytokine concentrations administered peripherally to be attained in the brain.

In addition, IL-2 exerts a modulatory action on DA release. In vitro studies report increases in the spontaneous and K^+^-stimulated DA release from rat striatal slices and mesencephalic cells in culture [99,100]. Equally in this case, in vivo studies reported conflicting findings. Specifically, the systemic administration of IL-2 was accompanied by reduced DA levels in the nucleus accumbens [101]. Differences in the concentration of this cytokine might be responsible for such discrepancies as suggested by Pettito and colleagues who observed a dose-dependent effect of IL-2 on DA release from striatal slices with excitation or inhibition by using low or high concentrations, respectively [102].

Reports on the effects of IFNs on brain catecholamines also describe an effect on DA levels. Chronic administration of IFN-α inhibits dopaminergic neuronal activity in the mouse brain and causes small decreases in whole brain DA or 3,4-dihydroxy-phenylacetic acid (DOPAC) [103]. Similarly, chronic treatment of monkeys with IFN-α causes a reduction in the concentration of DA metabolites homovanillic acid (HVA) and DOPAC measured in the CSF [104,105]. Such modification in DA content was paralleled by behavioral abnormalities such as a reduced locomotor activity and effort-based sucrose consumption, which suggest a decrease in striatal DA release [106] that could be reversed by the DA precursor levodopa (L-DOPA), implicating an effect of this molecule on DA synthesis and availability [107].

Conversely, Kumai and colleagues found that a chronic daily treatment of rats with IFN-α determines an increase in DA content in most of the brain regions analyzed [108]. However, again, differences in dosing and length of exposure may be responsible for these discrepancies.

Finally, a modulatory effect on DA release has been observed also for some chemokines. In particular, SDF-1α, also known as CXCL12, can stimulate DA release from nigral neurons [109]. As confirmed by current-clamp experiments, such a modulatory effect of DA neuron excitability is mediated by both indirect mechanisms, involving the release of glutamate and GABA onto DA neurons [110], and by direct potentiation of high voltage activated calcium (HVA-CA^2+^) currents [111]. Interestingly, although with a different mechanism of action, also the chemokine CCL2 has a neuromodulatory effect on DA neurons. The administration of this molecule through unilateral injection in the rat SN causes an enhancement of DA neuron excitability mainly due to the closure of K^+^ background channels. This increase in the membrane resistance triggers a boost of DA in the ipsilateral striatum and impacts the related locomotor activities.

Despite a direct modulatory effect on DA transmission has been demonstrated only for few cytokines, many others might affect DA release. Thus, the DA response associated with the immune stress caused by SARS-CoV-2 infection is actually a pattern of responses triggered by many cytokines and chemokines which have significant implications on DA-dependent behaviors.

### 3.4. Mechanisms Underlying the Cytokine-Induced Alterations in DA Release

DA concentration at the synaptic cleft is modulated at the level of synthesis, release, degradation, and reuptake. During SARS-CoV-2 infection the molecular machinery involved in these processes is perturbed following excessive stimulation by cytokines with significant alterations in neurotransmitter availability.

Proinflammatory cytokines (IL-1β, IFN-γ, IL-6, TNF-α) have been demonstrated to affect DA neurotransmission by decreasing the transport of tyrosine across the cell membrane in human fibroblasts [112]. Notably, the combination of some cytokines (i.e., IL-6 and TNF-α) causes a potentiated decrease in the activity of the tyrosine transporter suggesting the existence of a synergistic effect [112]. Such impairment in the uptake of the DA precursor, likely dependent on an alteration in the functionality/expression pattern of membrane bound _L_-type amino acid transporter 1 (LAT1) expressed at the BBB, represents a possible mechanism responsible for the alteration of DA synthesis induced by cytokines.

Most of the tyrosine, however, originates from the neuronal conversion of phenylalanine mediated by phenylalanine hydroxylase (PAH). Such enzyme, similarly to TH, requires the cofactor BH4 to perform the reaction of tyrosine generation. A decrease in the availability of BH4 is present in animal models following exposure to IFN-α [113] and those decreased BH4 concentrations are thought responsible for the impairment in DA neurotransmission [114] (Figure 1). Of note, a similar inhibitory effect on BH4 availability has been observed also for IL-6 in sympathetic neurons [115].

These observations are confirmed also in human studies where reduced BH4 activity, coupled to enhancement in IL-6 CSF levels, has been found in IFN-α-treated patients. In these people, the finding of a decreased CSF DA and its metabolite HVA, which correlate with an increased phenylalanine/tyrosine ratio, confirms a detrimental effect of these cytokines on DA transmission that mainly depends on alterations in DA synthesis [116,117].

IL-1 and TNF-α, in addition to reduce DA availability, seem to interfere with DA synaptic transmission by altering the DA packaging process. Both these cytokines are able to decrease the expression of VMAT2 [118] responsible for the loading of cytoplasmic DA into vesicles (Figure 1). This ability cannot be ruled out for other cytokines implicated in the changes of DA levels.

Some evidence exists indicating that cytokines may also target the DA reuptake mechanisms. At the current stage, the alteration of DA reuptake has been suggested to be mediated by the MAPK pathway [114] that is activated by IFN-α and other cytokines. This speculation is based on the fact that the constitutive expression of MAPK kinase (MEK) in DAT-expressing cells causes an increase in the maximal rate of DA uptake. Accordingly, the treatment of striatal synaptosomes with MEK inhibitors induces a decrease in DA reuptake [119].

Finally, DA signaling alteration by cytokines may be dependent on an impairment in DA receptor expression and function. For example, monkeys chronically treated with IFN-α present decreased D2 receptor binding in striatum using [^11^C]raclopride PET [106]. D2 autoreceptors are implicated in DA release regulation and thus changes in their expression levels following cytokine exposure may significantly alter DA neurotransmission.

Collectively, these studies provide evidence of a regulation of DA signaling by cytokines at different stages (Figure 1) and suggest that cytokines, released during SARS-CoV-2 infection, by acting synergistically and affecting several steps of synaptic transmission, may exert a significant impact on DA-dependent behaviors.

### 3.5. Glia Activation and Modulation of DA Release

Non-excitable cells in the CNS include astrocytes, oligodendrocytes, and microglia. Postmortem human brains of people who experienced SARS-CoV-2 infection have been found enriched for astrocytic proteins more than oligodendrocytes, neurons, or Schwann cells, suggesting that these are the most affected cells by SARS-CoV-2 infection and are responsible for mediating initiation and amplification of the neuroinflammation process [120].

In addition to the role of astrocytes in producing trophic factors and preserving neuronal functions by regulating extracellular pH through the removal of neurotransmitters, excess ions, free radicals, toxins, and debris, these cells play a key role in the CNS immune response. In particular, thanks to their ability to present antigens, they function as antigen-presenting cells to lymphocytes. In addition, they are important for activating other immune-responsive cells such as monocytes/macrophages/microglia. More importantly, in the context of the inflammatory responses, they are an abundant cellular source of immunoregulatory factors; in fact, they secrete and respond to a variety of both pro- and anti-inflammatory cytokines. Thus, in the multipartite synapse that astrocytes form with neurons, if on one side they provide support through neurotransmitter regulation and metabolic coupling, on the other side the massive cytokine production is responsible for induction or exacerbation of neuronal dysfunction.

Astrocytes have been implicated in DA transmission modulation. These cells seem to regulate [DA]_o_ by directly transporting and metabolizing DA [121] (Figure 2). This observation has been further confirmed by Adermark and colleagues [122] in an in vivo microdialysis study performed using the metabolic uncoupler fluorocitrate (FC), an astrocytes inhibitor. The application of FC shows increased striatal DA levels; conversely, it could be speculated that the overactivation of these cells, such as that occurring during SARS-CoV-2 infection, may cause an inhibition of DA release. One possible mechanism at the basis of changes in DA release triggered by astrocytic activation involves kynurenic acid, a neuroactive substance produced by astrocytes to facilitate the communication between the brain and the immune system [123]. Studies examining DA release have shown that kynurenic acid decreases striatal DA [124] (Figure 2). These data are confirmed by the fact that changes in kynurenine aminotransferase enzyme activity cause an enhancement in axonal DA release [125]. Alterations in DA levels are likely a consequence of a tonic inhibition over DA neurons induced by this metabolite; however, also modifications in striatal glutamate levels induced by kynurenic acid accumulation can indirectly modify DA release [126].

The glutamatergic control of striatal DA release has been largely demonstrated [127]. Glutamate exerts an inhibitory effect on DA release as shown by the increase in evoked DA measured in the presence of an AMPA receptor blocker [128]. However, due to the apparent absence of ionotropic glutamate receptors on DA axons [129], such influence is not directly dependent on glutamate-receptor activation. Rather, the effect of glutamate on DA release is indirect and requires the diffusible messenger hydrogen peroxide (H_2_O_2_), which suppresses axonal DA release via the activation of ATP-sensitive K^+^ (K_ATP_) channels [130] (Figure 2). Previous studies suggest that glutamate-dependent H_2_O_2_ must be generated in non-DA cells. Among striatal neuronal populations, SPNs have been identified as the primary cellular source of H_2_O_2_ [131] under physiological conditions. However, under a disease state such as during the SARS-CoV-2-induced inflammatory process, H_2_O_2_ may also originate from reactive astrocytes and directly inhibit DA release.

In addition to H_2_O_2_, excessive production of reactive oxygen species (ROS), including superoxide and the hydroxyl radical, occurs in activated astrocytes. In general, the increase in the generation of ROS is related to an impairment of mitochondrial function. In the case of SARS-CoV-2 infection, it has been proposed that the viral RNA localizes at the level of these organelles [132], thereby modulating their activity and causing not only an increase in ROS.

The ROS generated are implicated in the activation of microglia [133], the resident immune-competent cells of the CNS, which exert typical macrophagic functions such as phagocytosis, secretion of proinflammatory cytokines, and antigen presentation. Noteworthy, similar to astrocytes, microglial cells can respond to immunoregulatory factors, such IL-17, TNF-α, and IFN-γ, thereby exacerbating the neuroinflammatory process and the neuronal dysfunction.

During SARS-CoV-2 virus invasion, both activated microglia and astrocytes produce nitric oxide (NO) thanks to the inducible nitric oxide synthase (iNOS) enzyme that is expressed by these cells in response to inflammatory stimuli [134]. NO donors have been shown to enhance striatal DA release. Such a modulation of axonal DA release by NO can be direct or indirect via local circuits involving ACh and nicotinic ACh receptors (nAChRs) with a modification in the frequency dependence [135]. Moreover, in vivo observations suggest a NO-induced enhancement in DA levels via the inhibition of DAT transporter [136]. However, during SARS-CoV-2 infection, a detrimental effect of NO on DA neurotransmission probably dependent on high concentrations of NO (above nanomolar concentration) may be hypothesized. Indeed, NO reacts avidly with oxygen and superoxide radicals to form NO derivatives, including long half-life peroxynitrite. This powerful oxidant has been implicated as a causal factor in the inactivation of TH and, thus, in the DA synthesis failure [137] (Figure 2). Moreover, NO itself is able to increase the open probability of K_ATP_ [138]. Thus, it is possible that a mechanism similar to that used by H_2_O_2_ may occur and the release of DA is inhibited following a neuronal membrane hyperpolarization.

### 3.6. Upregulation of α-syn and Impairment of DA Release

Inflammatory mediators are known to contribute to neuronal impairment in part by triggering protein misfolding [139]. Disturbance in protein homeostasis, e.g., the cellular imbalance between the genesis of misfolded proteins and their degradation, can contribute to the accumulation of aggregated proteins [140] which are responsible for additional detrimental downstream events. SARS-CoV-2 has been found to be able to reshape the central cellular pathways comprising proteostasis [141], implying that the impairment in protein degradation might initiate a critical mass of aggregated proteins, including α-syn, which cause alterations in DA neurotransmission and cell injury.

α-syn has been identified as a nuclear and a presynaptic neuronal protein suggested to control vesicle fusion, neurotransmitter release, and synaptic plasticity [142,143]. In particular, α-syn regulates the activity of TH and AADC [144,145,146] and enhances the synaptic vesicle fusion and turnover [147,148,149,150], thereby modulating DA levels (Figure 1). Overexpression of α-syn is associated with a defect in DA synaptic transmission [151]. In vitro studies show that the reduction in TH promoter activity [152] and the inhibition of TH phosphorylation [153] following α-syn overexpression determine a decrease in DA levels, suggesting that such protein interferes with DA synthesis through the interaction with TH [144] (Figure 2). Accordingly, several mouse models overexpressing wild-type α-syn display reduced TH activity [154,155].

Furthermore, overexpression of α-syn also seems to decrease the rate of DA release [156,157]; this effect is probably due to a reduction in the vesicle recycling pool [158] or a defect in the exocytosis process [157,159].

Interestingly, beyond its synaptic function, α-syn is also thought to play a role in potentiating immune defense and is involved in the regulation of immune function. Indeed, engineered mice without syn genes (α, β and γ) exhibit deficient host defense against infectious agents [160,161]. In particular, α-syn knockout mice have a greater susceptibility and mortality than wild-type animals to viral and bacterial infections [162,163,164,165].

Cellular α-syn levels are increased during certain viral and bacterial infections. In fact, recent studies demonstrate an increase in α-syn expression after stimulation with IL-1 [166], supporting a role for α-syn in inflammation. Moreover, also the treatment with TNF-α seems responsible for increasing α-syn secretion and promoting cell-to-cell propagation of this protein [167]. Increased levels of α-syn are implicated in the recruitment of immune cells [164,168,169,170], activation of microglia, and induction of the NLR family pyrin domain containing 3 (NLRP3) inflammasome, a complex of intracellular proteins whose stimulation leads to the generation of proinflammatory cytokines with the goal of aiding the pathogen elimination [171,172,173,174].

Several reports have demonstrated the ability of the SARS-CoV-2 virus to bind and interact with α-syn [175,176]. Notably, this interaction seems to potentiate the activation of the NLRP3 inflammasome in microglia cells [177].

More importantly, the SARS-CoV-2 infection has been found responsible for causing an upregulation in α-syn expression and speeding up the aggregation process that triggers the formation of oligomers and amyloid fibrils typically observed in PD (Figure 2). This finding prompts the hypothesis that upregulation of α-syn during SARS-CoV-2 infection, or changes in α-syn dynamics, may cause changes in synaptic function and contribute to the alterations in DA release likely via an effect at TH or at the SNARE complex level which are implicated in DA synthesis and vesicle transport, respectively.

Intriguingly, both α-syn and SARS-CoV-2 spike protein have been found as a ligator for the toll-like receptor 4 (TLR4) protein [178,179]. In both cases, such interactions result in the activation of transcriptor factors encoding proinflammatory cytokines and IFNs. Thus, it can be speculated that together these two proteins amplify the inflammatory process, which further contributes to the DA signaling impairment.

## 4. Impact of Inflammation-Driven DA Dysregulation on PD Patients following SARS-CoV-2 Infection

The neuropathological hallmark of PD is the presence of intracellular inclusions, known as Lewy bodies (LBs), which are mainly composed of α-syn [180]. LBs are also found in extranigral neuronal populations (cortex, amygdala, locus coeruleus and peripheral autonomic system) where they contribute to promote neuronal dysfunctions that correlate with non-motor signs [181,182]. Despite the evidence of alteration in these extranigral sites, the nigrostriatal DA pathway is the main focus for research and therapeutic intervention in treating PD symptoms due to the key role of DA in the regulation of movement and motivational states.

A complex mix of aging, genetic susceptibility, and environmental insult, converging to varying degrees, underlies neurodegeneration occurring in PD [183,184,185].

While familial cases of PD are caused by genetic mutations [186], the majority of idiopathic PD (iPD) cases result from a complex interaction between genes and environment [187]. Several of the genes associated with PD risk, including the SNCA gene coding for α-syn, are involved in the immune system. Multiple lines of evidence both from studies performed in iPD patients’ brain and biofluids (CSF and serum) and in animal models of PD support the occurrence of a chronic inflammatory event that contributes to neuronal degeneration and symptomatology. The inflammation, occurring in a early stage of the disease, changes dynamically with disease progression and involves both peripheral immune cells and brain [188,189].

Sources of peripheral immune activation may include viruses and bacteria. In this regard, mounting epidemiological evidence links PD risk to immune disorders and certain types of infections [190]. Historical records reveal that the influenza A pandemic that occurred in 1918 was rapidly followed by an epidemic of encephalitis lethargica [191], an acute-onset polioencephalitis associated to post-encephalitic parkinsonism clinically characterized by bradikinesia, rigidity, postural instability, and resting tremors experienced by young adults patients months to years later [192,193]. Whether the 1918 influenza caused encephalitis letargica is still unclear; however, a recent study performed on formalin-fixed paraffin-embedded tissue samples shows no evidence for the presence of a specific or putative viral pathogen for post-encephalitic parkinsonism, suggesting a likely post-infectious immune-mediated etiology [194]. Interestingly, other viruses such as Coxsackie, Dengue, Epstein-Barr, hepatitis E, Japanese encephalitis B, measles, St. Louis encephalitis, and West Nile have been associated with transient or permanent parkinsonism [195]. Of note, the parkinsonian features observed in the context of these pathological conditions coexisted with encephalopathy [196,197,198,199,200]. Intriguingly, an association between PD and coronavirus infection has also been suggested. Recent data describe an exacerbation of PD symptomatology following SARS-CoV-2 infection [201]. Most interestingly, evidence of PD signs in previously undiagnosed people has been reported [9,10,61]. The frequency of encephalopathy in the context of COVID-19 seems to vary widely [202]; however, the close proximity between new-onset parkinsonisms and a COVID-19 diagnosis, and the presence of simultaneous encephalopathy in some patients, makes to hypothesize an etiological connection [9]. The cases also presenting encephalitis well fit the concept of general viral post-encephalitic parkinsonism described in the past for other encephalopaties [203]. Whether this post-encephalitic parkinsonism is characterized by the absence of α-syn pathology, such as in post-encephalitic parkinsonism arising following encephalitis letargica [204], is still unknown. The occurrence of a transient or permanent parkinsonism may be the result of a synergism between SARS-CoV-2-induced inflammation and a possible genetic predisposition to develop this neurodegenerative disorder.

The activation of the immune system and the release of proinflammatory molecules in the basal ganglia circuitry have largely been demonstrated to cause motor slowing and motivation reduction [105,205,206,207,208]. For instance, the administration of IFN-α, which induces the release of the inflammatory cytokines IL-6, IL-1 and TNF, causes depression, fatigue, motor slowing, and anhedonia [104,209,210]. Similarly, increased plasma levels of C-reactive protein and inflammatory cytokines correlate with both anhedonia and motor slowing [211,212], suggesting an association between increased inflammation and reduced motivation and motor functions. Intriguingly, IFN-α administration also causes an increase in glucose metabolism in the putamen [206,213]. Of note, the increased glucose metabolism in the basal ganglia is seen in PD [214,215,216] and seems to reflect the increased oscillatory burst activity secondary to the loss of inhibitory nigral DA input [217].

Studies performed to examine the effects of neuroinflammation on DA neuron activation clearly show that inflammatory molecules disturb DA signaling and DA-dependent phenomena. Acute administration of IFN-α to rhesus monkeys was found to decrease the rapid eye movement (REM) latency [218]. REM sleep is sensitive to DA and a similar observation has been drawn in PD [219].

Furthermore, a PET study conducted by Capuron and colleagues [207] shows that, in the caudate-putamen and ventral striatum, IFN-α causes increases in DA uptake and decreases in DA turnover. Although these data are in stark contrast to those observed in PD, where decreased uptake and increased turnover are seen, it should be noted that this observation has been performed in the presence of intact terminals. The decreased uptake in PD is thought to be a function of the loss of DA neurons and their projections, whereas the increased turnover indicates that the remaining neurons are capable of normal release [220,221]. Moreover, it is interesting to note that PD-like symptoms have been observed in patients treated with IFN-α and that these were responsive to levodopa, suggesting a reduced DA neurotransmission [222,223]. In light of this, it could be hypothesized that an increase in IFN-α levels following SARS-CoV-2 infection causes in the intact terminals a potential depletion of DA, which contributes to further worsen an already compromised scenario.

In this respect, we may hypothesize that SARS-CoV-2, through an inflammation-dependent impairment of DA neurotransmission, unmasks a latent neurodegenerative process and promotes the disease onset. There are obviously major gaps to fill to confirm this hypothesis. The early predegenerative changes in DA release, consisting in impaired transmitter release and reuptake and reduction in TH enzyme activity in the absence of any major cell loss, would represent an early defect that evolves in parallel with the development of a progressive degenerative process in the nigrostriatal axons and terminals. The first signs of behavioral impairments appear when the degeneration of DA neurons in SNc and the loss of TH-positive innervation in the striatum are present. Considering that the loss in SNc is specific to PD, we are aware that the absence of autopsy data in people infected with SARS-CoV-2 and recently diagnosed for PD does not allow reaching definitive conclusions on the mechanisms of the neuronal pathology and establishing a clear causal link between inflammation caused by COVID-19 and PD. Moreover, autopsy data would help to discriminate between PD and other diseases with parkinsonian symptoms that do not exhibit SNc death.

## 5. Conclusions

SARS-CoV-2 is endowed with the ability to infect midbrain neurons; whether the virus induces neuropathological alterations in people developing PD is still matter of debate. By contrast, it is evident that the virus induces a robust immune response characterized by a massive release of cytokines, chemokines, reactive oxygen species, and other inflammatory molecules that may affect DA neurotransmission.

Following SARS-CoV-2 infection, any change in [DA]_o_, caused by neuroinflammation, bears the potential to modulate basal ganglia output and cause symptoms reminiscent of PD. However, a possible direct viral effect within the substantia nigra is an important avenue to be explored too.

Indeed, a deeper understanding of the mechanisms by which SARS-CoV-2 infection disrupts cell viability and function is mandatory, in order to identify a potential target for therapeutic intervention.

## Figures and Tables

**Figure 1 ijms-24-05618-f001:**
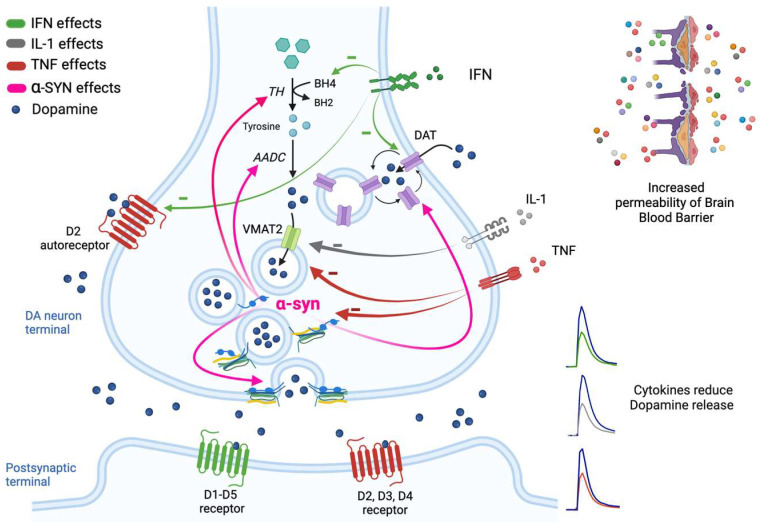
Potential mechanisms of cytokine effects on DA neurotransmission. Simplified representation. Cytokines released from peripheral immune cells or those produced in the brain by activated astrocytes and microglia lead to decreased concentrations of synaptic DA by effect on its synthesis and release. DA is synthesized in the cytoplasm by the action of tyrosine hydroxylase (TH) and amino acid decarboxylase (AADC). Once synthesized, DA is immediately sequestered into vesicles by the vesicular monoamine transporter 2 (VMAT2) and then released into the synaptic cleft where it binds to its postsynaptic receptors. DA signaling at synapse is terminated by degradation or reuptake of DA via the DA transporter (DAT). Inflammatory cytokines (e.g., IFN) may in fact impair the availability of DA by contributing to the oxidation of BH4, the cofactor required for the conversion of tyrosine to L-DOPA. They may also decrease the expression or function of VMAT2 (e.g., IL-1 and TNF) and/or increase the expression or function of DAT (e.g., IFN). The disruption in DA homeostasis induced by cytokines (e.g., TNF) may also be consequence of α-syn changes; in fact, α-syn is involved in the regulation of TH and AADC activity, in the regulation of synaptic vesicle fusion into the synaptic cleft, and in the trafficking of DAT to the cell surface. Finally, an inhibitory effect on release is also caused by a reduction in D2 receptors (e.g., IFN).

**Figure 2 ijms-24-05618-f002:**
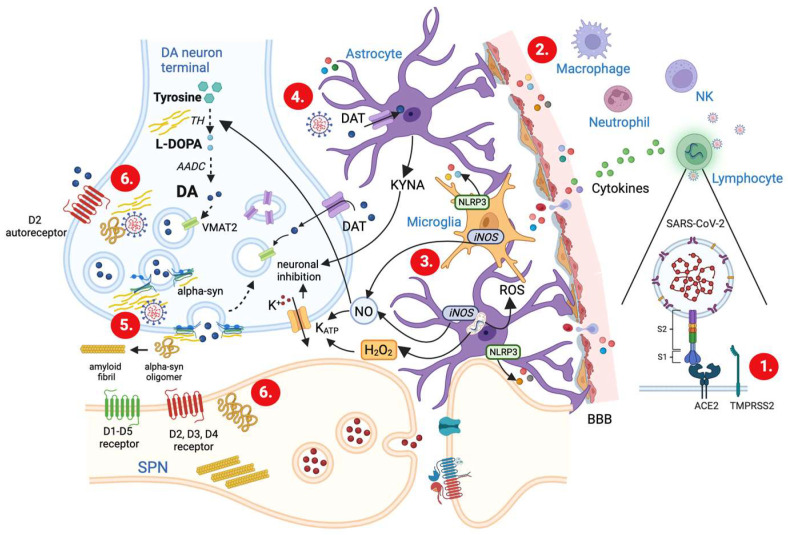
Schematic working model illustrating various proposed cellular mechanisms for how altered DA release could take place following SARS-CoV-2 infection. (1.) The spike protein of SARS-Cov-2 virus, by binding the protein angiotensin-converting enzyme 2 (ACE2), and under the participation of type II transmembrane serine protease receptors (TMPRSS2) [41], allows the virus to enter the cell and replicate itself using the RNA and protein synthesis machinery of the host. (2.) SARS-CoV-2 virus, via its spike protein, can activate peripheral immune cells (macrophages, T lymphocytes, NK cells, neutrophils), which release cytokines and chemokines. Cytokines, in turn, increase the permeability of the brain–blood barrier (BBB) and gain access to neurons, astrocytes, and microglial cells. (3.) Activated astrocytes and microglia synthesize other cytokines and produce many proinflammatory molecules, including hydrogen peroxide (H_2_O_2_), nitric oxide (NO), and kynurenic acid (KYNA), which cause neuronal inhibition and thereby reduce DA release. Also, microglia cells when activated express NLRP3 (NLR family pyrin domain containing 3) inflammasome, a complex of intracellular proteins which are involved in the maturation of proinflammatory cytokines (see text for details). (4.) The viral particles can enter the CNS (see text for details) and cause a direct activation of astrocytes and microglia that contribute to the inflammatory response in the CSF. (5.) The interaction between viral particles and α-syn causes impairment in vesicle docking and recycling and prevents the loading of newly synthesized and taken up DA into vesicle. (6.) After SARS-CoV-2 invasion, α-syn displays an increased trend to form oligomers and fibrils and to spread up throughout the synapse. Consequently, α-syn has an increased potential to affect DA transmission after viral invasion, compared to what was already described in Figure 1. Thus, the combined action of α-syn dysfunction and the increased inflammatory molecules could drive DA release alterations and participate in the onset of PD. CNS = Central Nervous System. CSF = Cerebral Spinal Fluid DAT = Dopamine Transporter. iNOS = inducible Nitric Oxide Synthase. ROS = Reactive Oxygen Species. SPN = Spiny Projection Neurons.

## Data Availability

No new data were created or analyzed in thi study. Data sharing is not applicable to this article.

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
