# Peer review of "Dopamine Transmission Imbalance in Neuroinflammation: Perspectives on Long-Term COVID-19"

_ijms, 2023, doi:10.3390/ijms24065618_

Round 1

Reviewer 1 Report

The manuscript presents a very detailed and comprehensive review, with a very well documented bibliography. So this is a very useful review. However, due to the complexity of the topic, we suggest that the two figures included be easier to understand.

1. In figure 1, potential mechanisms of cytokine effects on DA transmission, and the consequences for regulation in the nigrostriatal pathway, there are many abbreviations and could perhaps be better illustrated. The same occurs in Figure 2, which is intended to offer a schematic working model illustrating various proposed cellular mechanisms for how altered DA release could take place following SARS-CoV-2 infection.  

2. The final section of the manuscript, called the conclusion, is perhaps too long, and looks more like a summary. In a review article, more than a conclusion, what should be done is a recapitulation to highlight the main findings. In this sense, we suggest a modification.

Reviewer 2 Report

The title of the article is misleading: It suggests this is a study on a pathological basis on tissue studied affected by covid 19

The introduction did not quote any definitive studies or autopsy study that shows covid 19 affecti g dopamine is affected in production or loss of neurons in cell death.

Not sure what the authors are trying to state on the effects of covid 19 on the substantia niagra compacta.

This article is a good review of dopamine transmission but does not adequately support the title.

Also it does not adequately quote previous pathological studies that have addressed Parkinsonism and viral encephalitis .

The discussion is more review of pathology of dopamine release and less of how cells on a pathological basis are affected by viruses.

Would focus on autopsy studies and change the title as a review article.

Round 2

Reviewer 2 Report

Recommend including papers on the encephalitis lethargica epidemic and parkinsons disease

also in methodology state limitations of the theory as there is some suggestion of parkinsonism but not idiopathic parkinsons disease.

Also state the paper is a metaanalysis or review

Good papers state limitations 

may mention some limitations of the paper

it is a good paper and has good illustrations 

just few points to clarify then publish from my perspective 

parkinsonism should be differentiated from parkinsons disease.
